# Trends in COVID-19 case-fatality rates in Brazilian public hospitals: A longitudinal cohort of 398,063 hospital admissions from 1st March to 3rd October 2020

Ivan Ricardo Zimmermann[1]*, Mauro Niskier Sanchez[1], Gustavo Saraiva Frio[1], Layana Costa Alves[1,2], Claudia Cristina de Aguiar Pereira[3], Rodrigo Tobias de Sousa Lima[4], Carla Machado[5], Leonor Maria Pacheco Santos[1], Everton Nunes da Silva[1,6]

1 Department of Collective Health, Faculty of Health Sciences, University of Brasilia, Brasilia, Brazil,
2 Institute of Collective Health (ISC) at the Federal University of Bahia, Salvador, Brazil, 3 Oswaldo Cruz Foundation, National School of Public Health, Rio de Janeiro, Brazil, 4 Oswaldo Cruz Foundation, Leônidas e Maria Deane Institute, Manaus, Brazil, 5 Department of Preventive and Social Medicine, Faculty of Medicine, Federal University of Minas Gerais, Belo Horizonte, Brazil, 6 Collective Health Course, Faculty of Ceilândia, University of Brasilia, Brasilia, Brazil

* ivan.zimmermann@unb.br

**Data Availability Statement:** The programming code, dictionary and deidentified admissions data used in this study can be found at a public

## Abstract

### Background

Almost 200,000 deaths from COVID-19 were reported in Brazil in 2020. The case fatality rate of a new infectious disease can vary by different risk factors and over time. We analysed the trends and associated factors of COVID-19 case fatality rates in Brazilian public hospital admissions during the first wave of the pandemic.

### Methods

A retrospective cohort of all COVID-19-related admissions between epidemiological weeks 10–40 in the Brazilian Public Health System (SUS) was delimited from available reimbursement records. Smoothing time series and survival analyses were conducted to evaluate the trends of hospital case fatality rates (CFR) and the probability of death according to factors such as sex, age, ethnicity, comorbidities, length of stay and ICU use.

### Results

With 398,063 admissions and 86,452 (21.7%) deaths, the overall age-standardized hospital CFR trend decreased throughout the period, varying from 31.8% (95%CI: 31.2 to 32.5%) in week 10 to 18.2% (95%CI: 17.6 to 18.8%) in week 40. This decreasing trend was observed in all sex, age, ethnic groups, length of stay and ICU admissions. Consistently, later admission (from July to September) was an independent protective factor. Patients 80+ year old had a hazard ratio of 8.18 (95% CI: 7.51 to 8.91). Ethnicity, comorbidities, and ICU need were also associated with the death risk. Although also decreasing, the CFR was always around 40–50% in people who needed an ICU admission.

repository (https://github.com/ivanzricardo/covid19_lethality).

**Funding:** This study was funded by the grant MCTIC/CNPq/FNDCT/MS/SCTIE/Decit N° 07/2020 - Research on COVID-19, its consequences, and other severe acute respiratory syndromes, under the coordination of LMPS. GSF and LCA received a research scholarship from CNPq during the study. The study sponsor had no role in the study design, data collection, data analysis, data interpretation or report writing. The authors had full access to all study data and were responsible for the decision to submit for publication.

**Competing interests:** The authors have declared that no competing interests exist.

## Conclusions

The overall hospital CFR of COVID-19 has decreased in Brazilian public hospitals during the first wave of the pandemic in 2020. Nevertheless, during the entire period, the CFR was still very high, suggesting the need for improving COVID-19 hospital care in Brazil.

## Introduction

Novel coronavirus disease (COVID-19) is the major global public health threat today. As of 26 March 2021, there were more than 124,535,520 confirmed cases and 2,738,876 deaths reported worldwide (https://covid19.who.int). In Brazil, almost 200,000 deaths from COVID-19 have been reported only in 2020. The infection fatality rate of COVID-19 across countries was estimated to be 0.68% (0.53%–0.82%) based on a systematic review and meta-analysis of published studies until June 16, 2020 [1]. Another systematic review and meta-analysis assessed the case-fatality rate (CFR) of patients with confirmed [2] COVID-19 in intensive care units, showing a CFR of 41.6% (34.0–49.7%). The authors also suggested that the reported mortality rates declined from above 50% in March 2020 to close to 40% in May 2020. However, both meta-analyses showed considerable heterogeneity, which may mean that observed differences in results from the included studies are not comparable.

During the first wave of infections, COVID-19 CFR appeared to fall at the hospital level as the pandemic progressed [3]. In England, the hospital CFR declined from 6.0% on April 2 to 1.5% on June 15, 2020 [4]. A national cohort study in England has also indicated that this trend remained after adjustment for patient demographics and comorbidities [5]. In Germany, COVID-19 CFR have reduced across all age groups. A larger decrease was observed in the ages 60–79, with an average close to 9% in March/April falling to 2% in July/August 2020 [6]. In the USA, adjusted mortality dropped from 25.6% (23.2–28.1) in March to 7.6% (2.5–17.8) in August 2020 in New York City [7]. Similar results seem to be observed in Singapore and the Netherlands [8].

Empirical evidence on hospital CFR over time is scarce and skewed towards high-income countries. Therefore, it is critical to also gather evidence based on routinely collected health data from upper middle-income countries. In this context, Brazil provides a unique opportunity to study trends in hospital CFR over time. First, Brazil has a universal health system, in which 75% of the population (158 million Brazilians) receives health care exclusively through the public system [9]. Second, there is large regional disparity in access to healthcare services and health outcomes, which likely worsens with the austerity economic policies recently introduced [10]. Finally, there are important public datasets available covering a large sample of the affected population, such as the hospitalization authorizations (AIH) database, covering the individual reimbursement records of all hospital admissions in the public health system (http://sihd.datasus.gov.br).

Although it is not clear when the end of the first wave of COVID-19 infections in Brazil has happened, an overlap between ongoing first wave and second wave is likely to exist due to its heterogeneous geography [11]. Nevertheless, after reaching its peak, there was a sustained trend of reduction in the number of new infections between epidemiological weeks 30 and 45, which converges to call it the first wave of COVID-19 in Brazil (https://covid.saude.gov.br). Thus, we aimed to investigate the trends in COVID-19 hospital case-fatality rate (CFR) in Brazilian public hospitals and fatality risk factors, such as sex, age, ethnicity, comorbidities, and COVID-19 severity, during the first epidemic period in 2020.

## Methods

### Study design

This is a retrospective cohort study based on reimbursement records of hospital admissions in the Brazilian Public Health System (*Sistema Único de Saúde*, SUS). The present report follows the RECORD (Reporting of Studies Conducted using Observational Routinely collected Data) statement, an extension of the STROBE (Strengthening the Reporting of Observational Studies in Epidemiology) guidelines [12].

### Study setting

SUS provides health care, free of charge at the point of service, to the entire Brazilian population, covering both ambulatory and hospital care. The reimbursement of hospitalizations by SUS budget is done through the hospital admission authorizations (*Autorização de Internação Hospitalar*, AIH), document that identify the patient and the services performed during the hospital stay. The AIH is generated at the time o of admission to public or private hospitals that provide services for SUS, and are sent monthly to the Ministry of Health to enable billing of services delivered. The AIH are grouped and managed through the Hospital Information System (*Sistema de Informações Hospitalares do SUS*, SIHSUS), an administrative system that supports planning, regulation and control. Besides, SIHSUS allows the assessment of hospital morbidity and mortality profile and the quality of health care offered to the population, providing elements to improve health policies.

### Participants

The study population was based on available hospital admission records. For this purpose, any admission including ICD-10 ("U071", "U072", "B972" or "B342") or medical procedure codes ("0802010296", "0802010300", "0802010318" or "0303010223") related to COVID-19 (code descriptions available at http://sigtap.datasus.gov.br) in the diagnosis, causes of death and treatment fields of the reimbursement records was identified and classified as "COVID-19 related" hospital admission, becoming part of our study population. Our data cover the admissions that occurred between the 10th and 40th epidemiological weeks (from March 1 to October 3, 2020) according to the Brazilian epidemiological calendar (S1 File).

### Data sources, access and cleaning methods

All analyses were based on hospitalization authorization (AIH), which is public data available at the SIHSUS repository (htftp://ftp.datasus.gov.br/dissemin/publicos/SIHSUS/200801_/Dados) until the end of January 2021. In the AIH database, each hospitalization receives a unique key called the AIH number. If necessary, duplicated AIH numbers were filtered, considering only the main hospitalization record. The available data were fully anonymized before we accessed them.

In addition to cleaning and manipulation process in R language, the data was accessed with *microdatasus* package [13]. The programming code, dictionary and deidentified admissions data used in this study can be found at a public repository (https://github.com/ivanzricardo/covid19_lethality).

### Variables

We considered each unique hospitalization-level data on the variables corresponding to patient characteristics (sex, age, ethnicity, and comorbidities), clinical severity (length of stay, use of ICU and occurrence of death), geographical location and epidemiological week of

admission (complete description available in S1 Table in S1 File). Ethnicity was based on patient self-declaration of race/color at the time of admission, which could be classified as: white, black, brown, yellow, native Brazilian or not informed. The death outcome was based on the discharge information field ("discharge due to death") available in the hospitalization records, thus covering only the in-hospital deaths. The selected comorbidities were obesity, bacterial infection, cancer, diabetes, cardiovascular disease, kidney failure, HIV and symptoms and signs involving the circulatory and respiratory systems (ICD-10 codes R00-R09).

## Statistical analysis

**Hospital case-fatality rate.** Considering the date of admission as a reference point, the hospital CFR was estimated for each epidemiological week based on the proportion between the number of COVID-19 related admission that evolved to death and the total number of COVID-19 admission in that week. In order to make group comparisons, the weekly hospital CRF estimates were standardized through direct method assuming the age pattern of the total number of hospital admissions in the entire period as the reference population. Then, smoothing time series plots were created to analyse the trends of the hospital case-fatality rate and its variation according to sex, age, ethnicity, length of hospital stay and ICU use. Smoothing was based on local polynomial regression fitting (*loess*) [14] method available in geom_smooth() R function, which fits a polynomial surface determined by one or more numerical predictors using local fitting (fit is made using points in each point neighbourhood, weighted by their distance). Finally, we also planned to run additional Spearman correlation analysis to evaluate the effect of local caseload against the hospital CFR.

**Survival analyses.** A survival analysis approach was used to estimate the probability of death during the study period and to assess the relative significance of its associated factors. A descriptive analysis was performed to describe the frequency distribution and mean time by selected variables. For the survival analysis, the dependent variable was time, in days, defined as the difference between the date of death during follow-up and the date of admission.

The selection of the independent sociodemographic variables was based on the information available in the literature and the findings obtained in the stratified hospital case-fatality rate analysis. Univariate and multivariate analyses with Cox regression models were performed to estimate each independent factor hazard ratio (HR) and its 95% confidence interval.

All analyses were performed with R software, version 4.0.0 (RStudio Team. RStudio: Integrated Development Environment for R. Boston: RStudio, PBC; 2020) and its interface RStudio, version 1.3.959 (RStudio Team. RStudio: Integrated Development Environment for R. Boston: RStudio, PBC; 2020) and the Stata software program, version 14 (StataCorp. 2019. Stata Statistical Software: Release 16. College Station, TX: StataCorp LLC.).

## Results

### Participants

Based on our selection criteria, between March 1 (start of epidemiological week 10) and October 3 (end of epidemiological week 40), the SIHSUS received 398,063 reimbursement authorizations classified as COVID-19-related hospital admissions. Of these, 86,452 (21.7%) had death as the outcome (Fig 1).

### Hospitalizations

Among the 398,063 hospital admissions, the number of men hospitalized surpassed that of women during the entire period, corresponding to a proportion of 55.5% of hospitalizations in

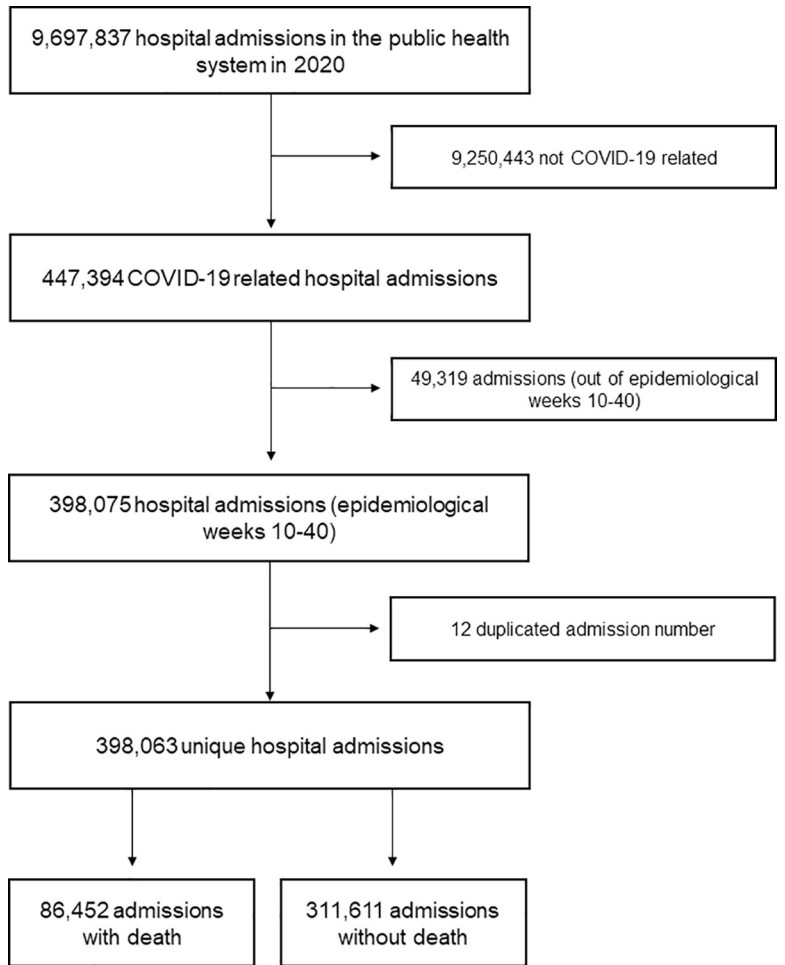

**Fig 1. Participant selection flowchart from the hospital admission records, Brazil, 2020.**

the entire study period (Table 1). During the entire period, 39.5% of admissions were of people between 60 and 79 years old, 31.1% of people between 40 and 59 years old, and 13.4% between people aged 80 and over. People aged between 20 and 39 years represented 12.5% of hospitalizations, and people aged 19 years or less accounted for 3.5% of hospitalizations. Black people represented 40.6% of hospitalizations in the entire period (brown 35.8%, black 4.8%), white 29.8%, Asian 4%. Native Brazilians (*indigenous*) represented only 0.3% of hospitalizations in the entire period, but approximately 25% of the observations did not contain information about the patient's ethnicity. Most hospitalizations lasted less than seven days (62.53%), followed by hospitalizations lasting between seven and 14 days (23.1%) and stays longer than 14 days (14.37%). Additionally, 26.07% of hospitalizations used the ICU.

Between epidemiological weeks 27 and 30, there was the highest number of hospitalizations: more than 20 thousand per week, with its peak being reached at the 28th week, when 21,461 hospitalizations were registered (Fig 2). As of 7/8/2020 (3,369 hospitalizations), a decreasing trend in the number of hospital admissions was observed.

Regarding geographic region, it was possible to observe the different times of the epidemics according to the local number of hospitalizations (S1 Fig). The Southeast region had the highest number of hospitalizations (172,084 admissions; 1.93 admissions per 1,000 inhabitants)

**Table 1. Characterization of 398,063 COVID-19-related hospital admissions, Brazil, 1ˢᵗ March to 3ʳᵈ October 2020.**

| Variable | Death outcome | | Total n (%) | |
|---|---|---|---|---|
| | **Yes n (%)** | **No n (%)** | | **p-value*** |
| Sex | | | | < 0.001 |
| Male | 50,137 (22.70%) | 170,692 (77.30%) | 220,836 (55.48%) | |
| Female | 36,315 (20.49%) | 140,919 (79.51%) | 177,239 (44.52%) | |
| Age | | | | < 0.001 |
| ≤ 19 years | 588 (4.19%) | 13,441 (95.81%) | 14,029 (3.52%) | |
| 20–39 year | 3,757 (7.58%) | 45,826 (92.42%) | 49,586 (12.46%) | |
| 40–59 years | 17,710 (14.3%) | 106,178 (85.70%) | 123,89 (31.12%) | |
| 60–79 years | 43,323 (27.58%) | 113,784 (72.42%) | 157,113 (39.47%) | |
| 80 + years | 21,074 (39.42%) | 32,382 (60.58%) | 53,457 (13.43%) | |
| Ethnicity | | | | < 0.001 |
| White | 25,206 (21.83%) | 90,276 (78.17%) | 115,482 (29.01%) | |
| Brown | 30,946 (21.69%) | 111,760 (78.31%) | 142,706 (35.85%) | |
| Black | 4,976 (26.19%) | 14,024 (73.81%) | 19,000 (4.77%) | |
| Asian descent | 3,027 (18.87%) | 13,014 (81.13%) | 16,041 (4.03%) | |
| Native Brazilians (indigenous) | 195 (18.82%) | 841 (81.18%) | 1,036 (0.26%) | |
| Not declared | 22,103 (21.29%) | 81,707 (78.71%) | 103,810(26.08%) | |
| Main diagnosis | | | | < 0.001 |
| Other disease | 5,472 (17.98%) | 24,967 (82.02%) | 30,439 (7.65%) | |
| COVID-19 | 80,981 (22.03%) | 286,655 (77.97%) | 367,636 (92.35%) | |
| High complexity admission | | | | 0,22 |
| No | 86,243 (21.72%) | 310,791 (78.28%) | 397,034 (99.74%) | |
| Yes | 210 (20.17%) | 831 (79.83%) | 1,041 (0.26%) | |
| ICU utilization | | | | < 0.001 |
| No | 36,378 (12.36%) | 257,932 (87.64%) | 294,310 (73.93%) | |
| Yes | 50,075 (48.26%) | 53,690 (51.74%) | 103,765 (26.07%) | |
| Time of admission | | | | < 0.001 |
| March | 424 (27.64%) | 1,110 (72.36%) | 1,534 (0.39%) | |
| April | 7,547 (29.38%) | 18,144 (70.62%) | 25,691(6.45%) | |
| May | 17,794 (25.37%) | 52,335 (74.63%) | 70,129 (17.62%) | |
| June | 17,562 (21.70%) | 63,376 (78.30%) | 80,938 (20.33%) | |
| July | 18,804 (20.35%) | 73,607 (79.65%) | 92,411 (23.21%) | |
| August | 14,304 (19.40%) | 59,416 (80.60%) | 73,720 (18.52%) | |
| September | 9,178 (18.56%) | 40,280 (81.44%) | 49,458 (12.42%) | |
| October | 840 (20.03%) | 3,354 (79.97%) | 4,194 (1.05%) | |
| Comorbidity | | | | < 0.001 |
| Not reported | 70,841 (20.08%) | 281,999 (79.92%) | 352,840 (88.64%) | |
| Cardiovascular disease | 7,368 (33.96%) | 14,326 (66.04%) | 21,964 (5.45%) | |
| ICD R000-R099 ** | 2,979 (27.77%) | 7,750 (72.23%) | 10,729 (2.70%) | |
| Diabetes | 3,299 (33.06%) | 6,681 (66.94%) | 9,980 (2.51%) | |
| Bacterial infection | 3,498 (65.67%) | 1,829 (34.33%) | 5,327 (1.34%) | |
| Respiratory disease | 958 (22.45%) | 3,309 (77.55%) | 4,267 (1.07%) | |
| Kidney failure | 2,680 (67.03%) | 1,318 (32.97%) | 3,998 (1.00%) | |
| Obesity | 851 (29.87%) | 1,998 (70.13%) | 2,849 (0.72%) | |
| Cancer | 636 (41.95%) | 880 (58.05%) | 1,516 (0.38%) | |
| HIV | 123 (26.34%) | 344 (73.66%) | 467 (0.12%) | |

(*Continued*)

**Table 1.** (Continued)

| Variable | Death outcome | | Total n (%) | |
|---|---|---|---|---|
| | **Yes n (%)** | **No n (%)** | | **p-value*** |
| Total | 86,453 (21.72%) | 311,622 (78.28%) | 398,063 (100%) | |

Notes

* Pearson's Chi-squared test

** Signs and symptoms relating to the circulatory and respiratory systems.

and the highest peak, reaching 9,051 admissions during the 28th week, but the Northern region had the highest rate (40,027 admissions; 2.14 admissions per 1,000 inhabitants). The Northern region was also the first to reach its admission peak during the 21st epidemiologic week with 2,654 admissions. The Midwestern region had the lowest number of hospitalizations and peak, with a total of 32,929 admissions (2.00 admissions per 1,000 inhabitants) and not surpassing 2,500 admissions during the entire analysed period, but the South region had the lowest rate (45,269 admissions; 1.50 admissions per 1,000 inhabitants). The Northeast region had a total of 107,754 admissions (a rate of 1.88 admissions per 1,000 inhabitants)

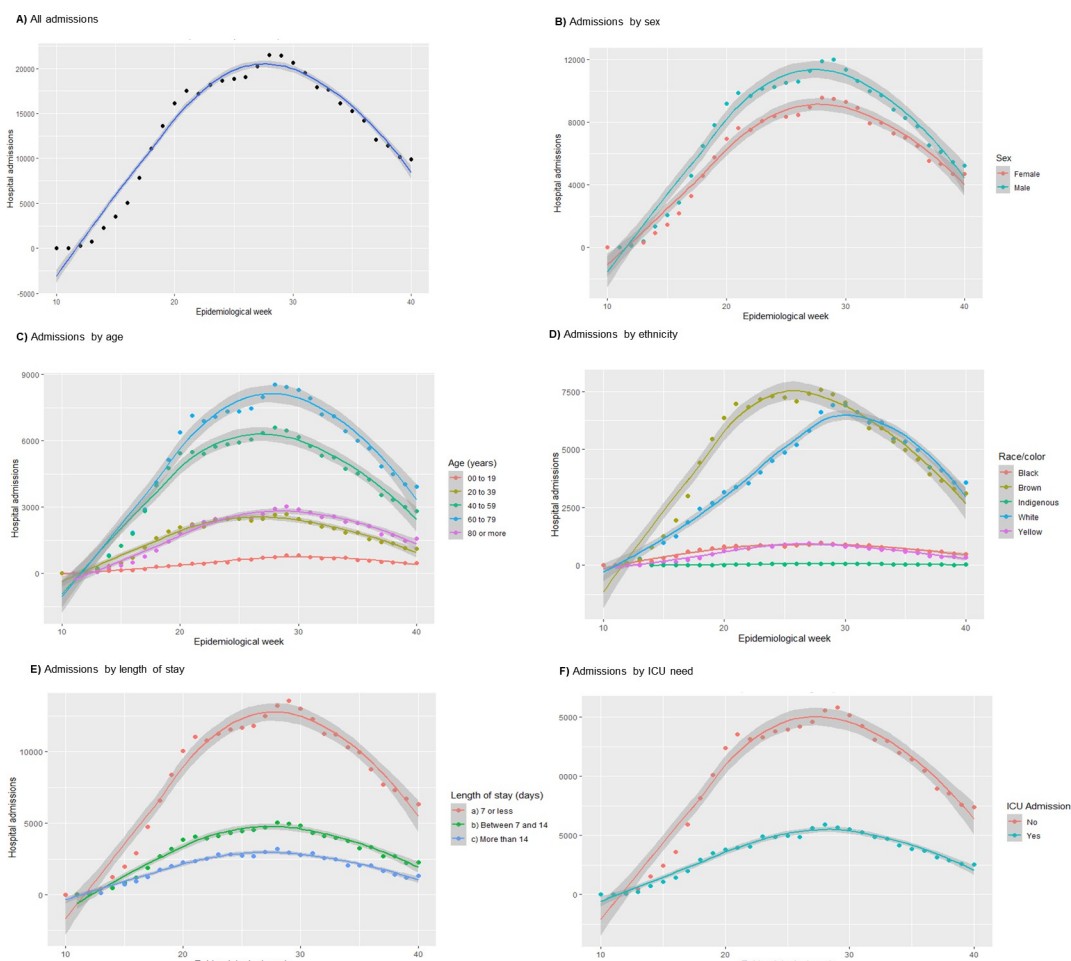

**Fig 2. Timeline of 398,063 COVID-19-related hospital admissions stratified by sex, age, comorbidities, ethnicity, length of stay and ICU need during epidemiological weeks 10 to 40, Brazil, 2020.**

## Hospital case-fatality rate (CFR)

Globally, after an initial growth trend until the 15th epidemiological week, hospital CFR trend decreased over time, varying from 31.8% (95%CI: 31.2 to 32.5%) in the 10th week to 18.2% (95%CI: 17.6 to 18.8%) in 40th week (Fig 3). This reduction was observed in both sexes, all age

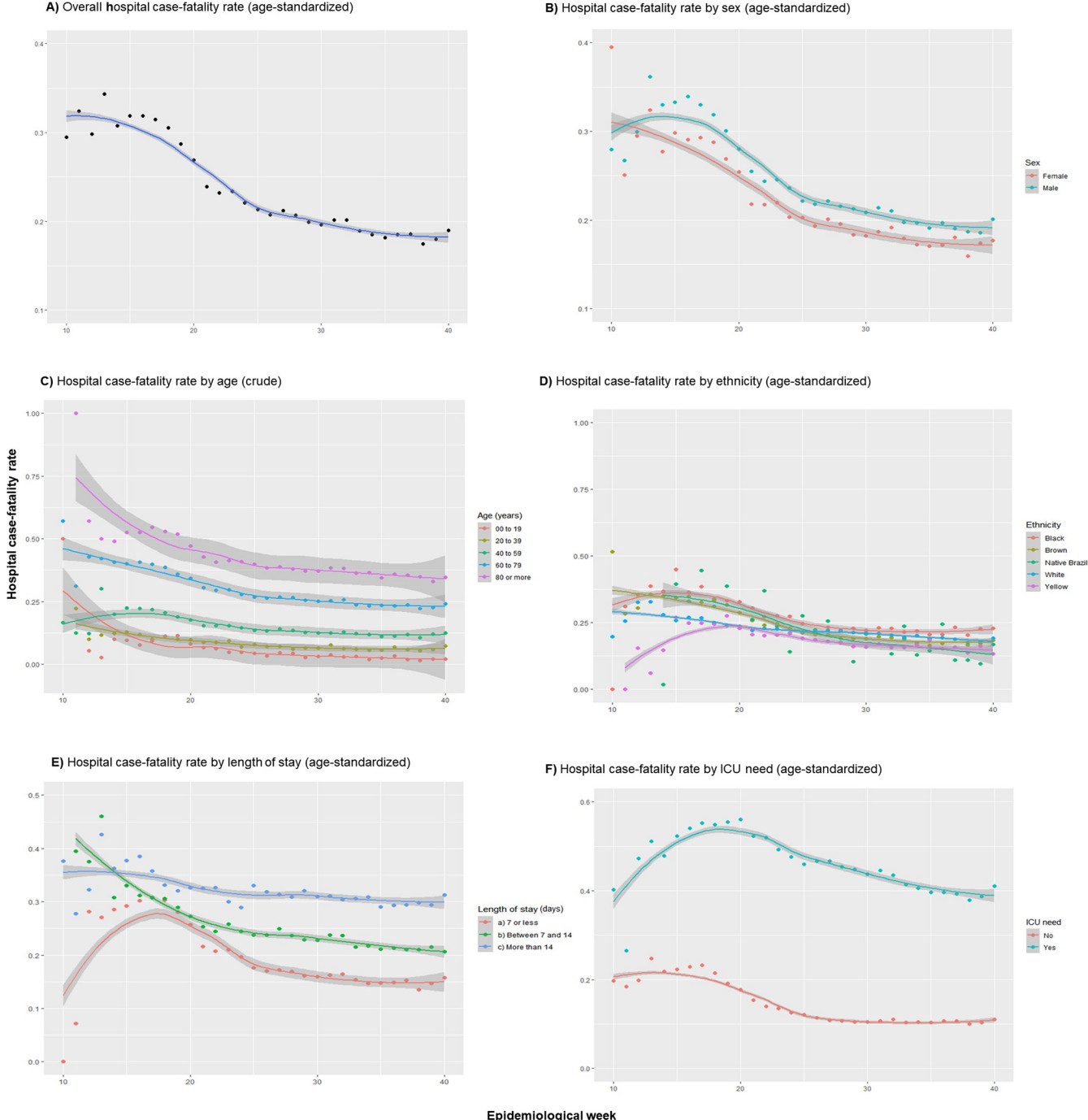

**Fig 3. Timeline of 398,063 COVID-19-related hospital case-fatality rates of all admissions stratified by sex, age, comorbidities, ethnicity, length of stay and ICU need during epidemiological weeks 10 to 40, Brazil, 2020.**

and ethnic groups. Proportionally, men died more than women, and age was directly proportional to the hospital CFR during the time. Black people data showed a higher hospital CFR and took longer to confirm reductions. In the 40th week, the age-standardized hospital CFR trend estimates were 22.53% (95%CI: 20.62 to 24.45%) in black, 18.06% (95%CI: 17.06 to 19.05%) in white, 17.78% (95%CI: 16.97 to 18.58%) in brown, 13.07% (95%CI: 09.27 to 16.86%) in native Brazilians and 14.69% (95%CI: 13.12 to 16.24%) in Asian people (complete hospital CFR data available in S2 Table in S1 File). However, as shown previously in Table 1, 26.1% of the ethnicity data were not available.

According to different lengths of hospitalization, it was also possible to see a hospital CFR decrease, but among those with up to seven days in duration. In this group, hospital CFR stands out, with a peak observed in the 18th epidemiological week. In addition, trends in age-standardized hospital CFR were clearly higher among people who were admitted to the ICU, reaching a peak in the 20th week of 41.03%. In the 40th epidemiological week, age-standardized hospital CFR trends were 38.40% (95%CI: 37.43 to 40.36%) and 10.91% (95%CI: 10.26 to 11.56%) in people who needed and who did not need an ICU admission, respectively.

The decreasing trend in hospital CFR was present in each of the 27 Brazilian states and it was clear in the North, Northeast and Southeast regions during all periods (S1 Fig). Nevertheless, this trend was not clear in the midwestern and southern regions, where the age-standardized hospital CFR started to decrease only after the 27th and 31st epidemiologic weeks, respectively. A complete pattern of the age-standardized hospital CFR during the epidemiological weeks in all of the 27 Brazilian states can be found in S2 Fig. In addition, as presented in S3 Fig, it was observed a strong negative correlation between the number of admissions per week and the age-standardized hospital CFR in the South (Spearman coefficient: -0.78; p-value < 0.001) and Southeast regions (Spearman coefficient: -0.79; p-value < 0.001).

## Survival analyses

Table 2 shows the results of the Cox regression model for fatality in Brazilian COVID-19-related hospital admissions. The adjusted model takes into account personal characteristics, as well as the region. Age was by far the most important individual hazard factor among those analysed, where those 60+ and 80+ years old presented a 4.7 and 8.1 increased likelihood of death, respectively. With a small but significant effect, women were less susceptible to die from COVID-19. The results indicate that compared to whites, any other ethnic group is more likely to die, except for individuals of Asian descent. Some comorbidities, such as obesity, diabetes and respiratory diseases, did not have a significant effect on fatality from the new coronavirus. The presence of other diseases and infections increased the probability of hospital case-fatality, with HIV showing the higher coefficient associated with death (HR: 1.36; 95% CI: 1.134–1.631).

Patients treated with procedures classified as high complexity were less likely to die (HR: 0.557; 95% CI: 0.485–0.639). On the other hand, patients who needed an ICU during admission had a twofold risk of a fatal outcome. Consistent with our previous timeline hospital CFR analysis, mortality also decreased from March to September (reaching a significant drop from July to September) in the survival analysis. There was a small and not significant increase in October.

## Discussion

This study has shown that after an early growth trend, the overall hospital case-fatality rate (CFR) of COVID-19-related admissions in Brazilian public hospitals decreased during the first

**Table 2. Hazard ratios for COVID-19 hospital mortality adjusted for exposure factors in the multivariate Cox regression, Brazil, 1st March to 3rd October 2020.**

| Variable/Category | Crude values | | Adjusted values | |
|---|---|---|---|---|
| | HR | 95% CI | HR | 95% CI |
| Sex | | | | |
| Male | 1.000 | | 1.000 | |
| Female | 0.929** | 0.916–0.942 | 0.923** | 0.910–0.936 |
| Age | | | | |
| ≤ 19 years | 1.000 | | 1.000 | |
| 20–39 year | 1.887** | 1.725–2.064 | 1.978** | 1.807–2.165 |
| 40–59 years | 2.889** | 2.653–3.145 | 2.939** | 2.697–3.202 |
| 60–79 years | 4.765** | 4.380–5.185 | 4.787** | 4.397–5.213 |
| 80 + years | 7.658** | 7.034–8.338 | 8.178** | 7.505–8.911 |
| Ethnicity | | | | |
| White | 1.000 | | 1.000 | |
| Brown | 1.076** | 1.058–1.094 | 1.045** | 1.023–1.067 |
| Black | 1.140** | 1.106–1.176 | 1.093** | 1.057–1.130 |
| Asian descent | 0.986 | 0.949–1.024 | 0.975 | 0.935–1.017 |
| Native Brazilians (indigenous) | 1.126 | 0.970–1.307 | 1.247** | 1.072–1.449 |
| Not declared | 1.093** | 1.073–1.113 | 1.107** | 1.083–1.132 |
| Main diagnosis | | | | |
| Other disease | 1.000 | | 1.000 | |
| COVID-19 | 1.190** | 1.157–1.224 | 1.080** | 1.046–1.115 |
| High complexity admission | | | | |
| No | 1.000 | | 1.000 | |
| Yes | 0.535** | 0.468–0.612 | 0.557** | 0.485–0.639 |
| ICU utilization | | | | |
| No | 1.000 | | 1.000 | |
| Yes | 1.985** | 1.958–2.014 | 2.077** | 2.046–2.109 |
| Time of admission | | | | |
| March | 1.000 | | 1.000 | |
| April | 1.180** | 1.064–1.310 | 1.131* | 1.013–1.263 |
| May | 1.166** | 1.052–1.292 | 1.073 | 0.963–1.197 |
| June | 1.004 | 0.906–1.112 | 0.912 | 0.818–1.017 |
| July | 0.961 | 0.868–1.065 | 0.881* | 0.791–0.983 |
| August | 0.934 | 0.842–1.034 | 0.851** | 0.763–0.949 |
| September | 0.913 | 0.823–1.012 | 0.831** | 0.744–0.927 |
| October | 0.970 | 0.859–1.095 | 0.884 | 0.779–1.004 |
| Comorbidity | | | | |
| Not reported | 1.000 | | 1.000 | |
| Cardiovascular disease | 1.155** | 1.127–1.183 | 1.048** | 1.017–1.081 |
| ICD R000-R099 * | 1.210** | 1.167–1.255 | 1.178** | 1.131–1.227 |
| Diabetes | 1.145** | 1.106–1.187 | 1.040 | 0.998–1.084 |
| Bacterial infection | 1.646** | 1.590–1.704 | 1.521** | 1.462–1.582 |
| Respiratory disease | 0.989 | 0.929–1.053 | 0.961 | 0.899–1.027 |
| Kidney failure | 1.457** | 1.401–1.516 | 1.178** | 1.128–1.231 |
| Obesity | 0.929* | 0.869–0.994 | 1.011 | 0.942–1.086 |
| Cancer | 1.310** | 1.210–1.418 | 1.398** | 1.278–1.529 |
| HIV | 0.757** | 0.632–0.906 | 1.360** | 1.134–1.631 |

(*Continued*)

**Table 2.** (Continued)

| Variable/Category | Crude values | | Adjusted values | |
|---|---|---|---|---|
| | HR | 95% CI | HR | 95% CI |
| Total number of observations | | | | 398,063 |

* p<0.05

** p<0.01

*** Signs and symptoms relating to the circulatory and respiratory systems.

wave period. This trend was associated with several factors, including age, sex, ethnicity, need for ICU and geographic region.

The downward trend we detected is in line with what most of the literature has been indicating recently. In a cohort of over 1,600 hospitalized patients in Spain admitted between March and September 2020, mortality decreased from 11.6% to 1.4% in the last month [15]. A similar period was analysed in the United States and indicated a drop in adjusted mortality from 25.6% in March to 7.6% in August [7]. In Germany, COVID-19 fatality rates have reduced across all age groups. A larger decrease was observed in the ages 60–79, with an average close to 9% in March/April falling to 2% in July/August 2020 [6]. The same pattern has been observed in England, where the hospital fatality ratio fell from 6% in early April to 1.5% in mid-June 2020 [4]. In France, time analysis of hospital CFR also showed a decrease over time, globally and in almost all districts [3]. Although the reasons for this reduction in hospital CFR are unknown since there is no specific treatment for COVID-19, some studies have raised potential reasons, such as a higher proportion of younger patients with fewer comorbidities over time; health workers have become more skilled at the management of severe patients during the epidemic; and early use of remdesivir to patients not requiring mechanical ventilation and dexamethasone to those requiring supplemental oxygen or on mechanical ventilation [3, 8, 15, 16]. Nevertheless, as much as Brazil has shown a decreasing trend as other countries, it is concerning that despite presenting an important relative drop (approximately 33%), it remained at a high level at week 40 –an age-standardized rate around 18%. It is worth noting that case fatality rates are not directly comparable among countries due to high heterogeneity in terms of countries' health information systems (in-hospital versus 30-day after discharge), quality of registry (mandatory versus voluntary report), disease classification (ICD-10 versus ICD-9 or other earlier versions), and completeness of the information.

In survival analysis, sociodemographic variables and some comorbidities were identified as associated with COVID-19 hospital mortality. Like what has been identified in other studies, in Brazil and other countries, males, increasing age, black and brown ethnic group, displayed higher adjusted hazard ratios of death. Blacks, browns, and native Brazilians (*indigenous*) people were more likely to die during hospitalization in all estimated models. Previous studies that used data from the Hospital Information System (SIM) [17] and Epidemiological Surveillance Information System (SIVEP-Gripe) [18] to study hospital mortality related to COVID-19 in Brazil in the first semester of 2020 showed that being black or brown was an important risk factor for hospital mortality. In other countries, higher in-hospital mortality for blacks in comparison to whites has also been observed [17–20]. In Brazil, such findings may be more a reflection of socioeconomic vulnerabilities to which blacks, browns and native Brazilians are exposed, which influence living conditions, lifestyles, access to healthcare, and ultimately have implications for their health, including COVID-19 outcomes [21]. In other countries, such as England and the United States, sociodemographic inequalities are also correlated with higher mortality among blacks and Hispanics when compared to whites [19, 22].

Different from other reports in the literature that looked at obesity, respiratory diseases, and diabetes [23, 24], our results were not significant in adjusted models on these comorbidities. This may be explained by the level of reporting of these variables in an administrative data source. Higher risks of mortality were observed for comorbidities such as cancer, bacterial infections, heart disease, other symptoms involving the circulatory and respiratory systems and kidney failure. A systematic review and meta-analysis conducted by Yang and colleagues (2021) [25] identified that patients with a cancer diagnosis were more susceptible to COVID-19 and were at increased odds of dying from COVID-19. Similar findings were observed in other studies, including meta-analyses for coronary heart disease and COVID-19 [26, 27]. A systematic review and meta-analysis carried out to evaluate the significance of demographics and comorbidities in COVID-19 demonstrated that metabolic diseases, comprising CVD, diabetes, hypertension and respiratory diseases (COPD and others), were the most common comorbidities associated with a severely poor prognosis and severe outcomes [28]. Between March and September 2020, a cross-sectional study carried out in 25 hospitals in the South and Southeast regions of Brazil concluded that the high risk of hospital mortality was associated with having hypertension, being male, ages over 69 years, having kidney disease and for patients who were admitted in the ICU, mortality was 47.6% [29]. For HIV, there is emerging evidence suggesting a moderately increased risk of COVID-19 mortality among people living with HIV (PLWH) [30].

## Strengths and weaknesses of this study

Using a rich dataset covering 398,063 hospital admissions for COVID-19 over a 7-month period, we provided estimates of the COVID-19 hospital CFR trends by epidemiological week at public hospitals in Brazil (stratified by sex, age, and ethnicity) and risk factors related to COVID-19 mortality (controlled for sex, age, ethnicity, comorbidities, month of hospital admission, type of hospital and impatient stay). To the best of our knowledge, this is the first study to investigate trends in COVID-19 hospital CFR in an upper middle-income country in a real-world reimbursement dataset of that size.

Some limitations of our study should also be acknowledged. First, in addition to a high level of missing information on ethnicity, epidemiological information on pre-existing comorbidities, have low quality and completeness in administrative datasets since this information is not mandatory for reimbursement. Based on this, our estimates from these variables must be interpreted with caution due to potential underreporting bias. Second, approximately 25% of the Brazilian population has private health insurance, and our findings may not reflect trends and risk factors related to COVID-19 in private hospitals. Third, although the reduction in the COVID-19 hospital death rate trends over time may be explained by improvements in clinical practice, we were not able to examine this causality effect due to a lack of clinical practice records at secondary information systems. Third, the COVID-19 outbreak was asymmetrically distributed across the country and over time, particularly in large cities; thus, all global trends should be interpreted with caution. Finally, our findings reflect the hospital CFR trends and mortality risk factors related to COVID-19 admissions in the first wave of the pandemic. Finally, we highlight that our data only covers in-hospital deaths reimbursed by SUS and should not be generalized to hospitalizations in private health insurance networks, paid out-of-pocket or deaths that occurred in other settings such as home deaths.

## Implications for clinical practice and health policy

Since the pandemic emerged, the main concern of health authorities worldwide has been the collapse of healthcare systems and the lack of hospital beds for patients with moderate and

severe COVID-19. Our findings suggest that the response of the Brazilian public health system (SUS) to the COVID-19 pandemic from March to October 2020 was able to achieve a sustained reduction in hospital CFR over time. However, there is a long way to go in terms of achieving stability, since the crude hospital CFR for patients with COVID-19 remained high in our data (approximately 20% in October 2020). There has been a large effort to provide ICUs and respirators to public hospitals but less attention to the training of health workers to support clinical practice and the management of ventilated patients [18], which directly impacts clinical outcomes. There are also concerns about geographical access to hospital beds and ICUs across Brazilian municipalities. A study suggested that the average distance travelled by patients from 464 Brazilian municipalities (8% of the total) was more than 240 km to obtain the ICU [31].

Our results also highlighted the population groups at higher risk of death due to COVID-19 at the hospital level: elderly people; native Brazilians (*indígenas*), patients with comorbidities, and hospitalized in the ICU. These population groups were prioritized by the National Immunization Programme in Brazil (except for patients hospitalized in the ICU) since vaccination against COVID-19 started in mid-January 2021. Finally, since December 2020, there has been a strong resurgence of new cases and deaths in Brazil (https://covid.saude.gov.br), as well as the emergence of a new SARS-CoV-2 variant [32], and it is important to state that our results could not reflect this new reality.

## Supporting information

**S1 Fig.** Timeline of 398,063 COVID-19-related hospital admissions (left) and age-standardized hospital case-fatality rates (right) stratified by geographic region during epidemiological weeks 10 to 40, Brazil, 2020.
(TIF)

**S2 Fig. Hospital case-fatality rates stratified by Brazilian states during epidemiological weeks 10 to 40, Brazil, 2020.**
(TIF)

**S3 Fig.** Correlation analysis between age-standardized hospital case-fatality rates and the number of hospital admissions per week stratified by A) All regions, B) North, C) Northeast, D) South, E) Southeast and F) Midwest region during epidemiological weeks 10 to 40, Brazil, 2020.
(TIF)

**S1 File.**
(DOCX)

## Author Contributions

**Conceptualization:** Ivan Ricardo Zimmermann, Mauro Niskier Sanchez, Gustavo Saraiva Frio, Leonor Maria Pacheco Santos, Everton Nunes da Silva.

**Data curation:** Ivan Ricardo Zimmermann, Gustavo Saraiva Frio, Layana Costa Alves.

**Formal analysis:** Ivan Ricardo Zimmermann, Gustavo Saraiva Frio, Layana Costa Alves.

**Funding acquisition:** Leonor Maria Pacheco Santos.

**Investigation:** Ivan Ricardo Zimmermann, Gustavo Saraiva Frio, Layana Costa Alves.

**Methodology:** Ivan Ricardo Zimmermann, Gustavo Saraiva Frio, Layana Costa Alves.

**Supervision:** Mauro Niskier Sanchez, Claudia Cristina de Aguiar Pereira, Rodrigo Tobias de Sousa Lima, Carla Machado, Leonor Maria Pacheco Santos, Everton Nunes da Silva.

**Validation:** Ivan Ricardo Zimmermann, Mauro Niskier Sanchez, Gustavo Saraiva Frio, Layana Costa Alves.

**Visualization:** Ivan Ricardo Zimmermann, Gustavo Saraiva Frio, Layana Costa Alves.

**Writing – original draft:** Ivan Ricardo Zimmermann, Mauro Niskier Sanchez, Gustavo Saraiva Frio, Layana Costa Alves, Claudia Cristina de Aguiar Pereira, Rodrigo Tobias de Sousa Lima, Carla Machado, Leonor Maria Pacheco Santos, Everton Nunes da Silva.

**Writing – review & editing:** Ivan Ricardo Zimmermann, Mauro Niskier Sanchez, Gustavo Saraiva Frio, Layana Costa Alves, Claudia Cristina de Aguiar Pereira, Rodrigo Tobias de Sousa Lima, Carla Machado, Leonor Maria Pacheco Santos, Everton Nunes da Silva.

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
