## [Decision Letter · Decision Letter 0]

14 May 2021

PONE-D-21-11421

Trends in COVID-19 case-fatality rates in Brazilian public hospitals: an analysis based on 398,063 hospital admissions records from 1st March to 3rd October 2020

PLOS ONE

Dear Dr. Zimmermann,

Thank you for submitting your manuscript to PLOS ONE. After careful consideration, we feel that it has merit but does not fully meet PLOS ONE’s publication criteria as it currently stands. Therefore, we invite you to submit a revised version of the manuscript that addresses the points raised during the review process.

We look forward to receiving your revised manuscript.

Kind regards,

Aleksandar R. Zivkovic

Academic Editor

PLOS ONE

Journal Requirements:

2) Please include captions for your Supporting Information files at the end of your manuscript, and update any in-text citations to match accordingly. Please see our Supporting Information guidelines for more information: http://journals.plos.org/plosone/s/supporting-information.

3) In your ethics statement in the Methods section and in the online submission form, please clarify whether all data were fully anonymized before you accessed them.

Reviewers' comments:

Reviewer #1: This is an interesting paper, and useful as a resource to examine the changing nature of in-hospital fatality for COVID-19 in Brazil. I hope that the authors edit the paper, as it would be a shame to see it go unpublished.

There are several major issues with the publication as it currently stands:

1. The paper reports adhering to the STROBE guidelines for observational research, but does not. Some of these issues are minor, such as the title which should properly identify the study as a longitudinal cohort according to STROBE, but some are more major. In particular, the covariates used in the primary analysis are not well described and are very hard to understand. How is ethnicity defined in hospital collections in Brazil? How is this data accessed? What comorbidities were included, how were they identified (ICD codes?), does this leave room for error and which errors if so? I would suggest the authors carefully go through not just the STROBE checklist, but the accompanying papers to ensure that they are indeed meeting the guidelines for reporting.

2. There is no information on how deaths were garnered from the records, which is a major weakness of the research. Were these in-hospital deaths? Does Brazil have a linked hospital/death reporting system? If these are in-hospital deaths, reporting lags may be a large issue - if not, death reporting across regions should be considered as a potential weakness.

3. One hypothesis for the changing hospital CFR in many places is the nature of the pandemic itself. In the UK, hospital CFRs fell from the peak until the second wave, and then increased again. This is potentially due to an overwhelming effect, whereby a large number of COVID-19 cases changes the type of patient who is admitted to hospital, and thus changes the denominator for the in-hospital CFR. While it may be impossible to fully examine the impact of such changes, it is clear from the trends in CFR when stratified by age and ethnicity that there is some impact. It might be useful for this study to look at the hospital CFR for each region over time against the current caseload of that area, although this is an addition that would take some extra work.

4. There is insufficient information in the text about missing data. While routine hospital data is incredibly useful, it is also usually filled with missing fields. While the dataset appears to be impressive, there should be a detailed discussion in the manuscript of how missing data was managed.

5. The statistical analysis is currently not fully described. The method of obtaining smoothed curves should be elucidated.

6. A somewhat minor point, but to me the tables are extremely hard to read. I would suggest having more columns and fewer rows, perhaps breaking each table down by age group. Similarly, the regression outputs are hard to read, especially given that the reference categories have been excluded.

7. While the introduction and discussion are good, I would ask for more information particularly in the introduction about the Brazilian hospital system. For international readers, there is scant detail on how it works - an additional paragraph would be very helpful to understand the context.

8. Currently, the link the authors have provided to their data/code goes to github's main webpage. Probably a typo.

Reviewer #2: This article describes the evolution of hospital lethality due to COVID-19 in the public hospital care system in Brazil. The article uses patient health data collected by the Brazilian public social insurance system, which covers more than 300,000 patients, with the indication of the duration of hospitalization and the outcome of the hospitalization. This large dataset allows a precise analysis of the evolution of hospital lethality and the article is of undeniable interest.

Some general remarks:

1. Brazil is a very big country and have a very heterogeneous geography. The authors could present in more detail the differences between the different States of Brazil (instead regions).

2. It would be interesting to have indication on the duration of the hospitalization (statistical distribution, relationship with the outcome of the disease, etc.).

3. To study the dynamics and evolution of CFR and to compare rates over time, if we want to exclude known factors related to death (mainly age), we have to standardize on age between weeks: therefore, we have to take into account the evolution of the age structure of patients over time.

4. In several countries a strong correlation between morbidity and case fatality rates has been observed. It would be interesting to have this analysis also for Brazil. And in particular to do it by state, because the differences between regions in the evolution of the disease are sensitive (but as mentioned before, when comparing geographical units, it is necessary to standardize on age).

5. Finally, it should be noted that it is always difficult to compare hospital case fatality rates between countries, because even within areas with comparable health systems (e.g., the EU), these hospital case fatality rates show differences that cannot be explained solely by differences in patient management, but first by the difference between countries in definition, declaration and reporting systems for morbidity and mortality

6. The quality of illustration can be improved. Maps are welcome.

Some minor remarks:

1. Why calculate the lethality rate per week, when the calculation can be done per day and then smoothed per week?

2. Line 58, 59. Hospital (or inpatient, or intensive care unit inpatient) fatality rate (not mortality rate)

3. Line 63-71: need to adjust the terminology (death rates, hospital fatality rate, adjusted mortality…). I think that « inpatient case-fatality rate » or « hospital case-fatality rate » is appropriate.

4. Line 142: better “proportion between the number of COVID-19 related admission that evolved to death and the total number of COVID-19 admission in that week”.

5. Line 170-179: all these results by category (age, ethnicity, comorb.) in the hospitalized COVID-19 population must be compared with the proportion of the same category in the global population, and the authors must indicate if the differences are statistically significant. Table 1 must present these results.

6. Line 182-184: Table 1 is mixing different information. I think it must be splitled in various tables. For example, Comorbidity analysis with CFR differences will be very interesting.

7. Line 195-201: Must present morbidity rates and not only morbidity.

8. Line 208-212: comparison of CFR between ethnic group is interesting only if major cause of death (age) in excluded: data must be standardized on age before comparison. Also, size of groups is different: it is needed to present confidence intervals.

9. Line 223-226: Is there a relationship between morbidity and CFR, as observed in other countries? Globally? By region? It is possible that changes in CFR are directly related to morbidity, so the study should be refined by analyzing data on regions (or better on states).

10. Line 235-239: this is very strange. Perhaps the explanations given in the discussion (data quality) could be the subject of an earlier paragraph in the data and methods section.

11. Line 246: there is no indication of the source of the data on the health care system or the criteria used to characterize it ("well-equipped and staffed hospital" is not enough).

12. Line 266: reference needed.

13. Line 336: remark: the same level of CFR was observed in France in second and third wave.

Reviewer #3: This is a straightforward epidemiological study of a cohort of nationwide COVID-admissions in Brazil from March to October 2020 analyzing in-hospital mortality and its associated factors.

The study has been well planned and executed. The sheer size of the cohort gives the opportunity to examine factors influencing the CFR with a high power.

I would wish the authors to clarify only a couple of points:

The decrease in CFR is highest and most pronounced in the short term hospitalized group. This raises the question, whether the populations hospitalized in different categories (short term, non-ICU vs. ICU) changed over time. Examining these questions could be important for the interpretation of the time trends shown.

In addition, if there are changes in the populations an additional analysis stratified for variables with clear changes over time should be done (or reported if already done).

6. PLOS authors have the option to publish the peer review history of their article (what does this mean?). If published, this will include your full peer review and any attached files.

Reviewer #1: **Yes: **Gideon Meyerowitz-Katz

Reviewer #2: No

Reviewer #3: **Yes: **Bernd Salzberger, MD

---

## [Author Response · Author response to Decision Letter 0]

29 Jun 2021

Dear Editor,

We carefully considered each of the reviewers’ suggestions. As a result, it is our opinion that the reviewers made an exceptionally good contribution to the clarity and the quality of our paper.

Please find below all the reviewers’ comments with our responses and respective indications of changes in the manuscript:

No. Reviewer’s #1 Comments Authors’ Answers

-- This is an interesting paper, and useful as a resource to examine the changing nature of in-hospital fatality for COVID-19 in Brazil. I hope that the authors edit the paper, as it would be a shame to see it go unpublished ANSWER: Thank you very much. We carefully considered each of the suggestions to be able to publish the paper.

1 The paper reports adhering to the STROBE guidelines for observational research, but does not. Some of these issues are minor, such as the title which should properly identify the study as a longitudinal cohort according to STROBE, but some are more major. ANSWER: We have revised our manuscript in order to be consistent with RECORD statements checklist (http://record-statement.org), including our new title suggestion as:

Trends in COVID-19 case-fatality rates in Brazilian public hospitals: a longitudinal cohort of 398,063 hospital admissions from 1st March to 3rd October 2020

We would like to emphasize that even RECORD extension including specific statements for studies using routinely-collected health data (such as health administrative data), like statement 6.1 “The methods of study population selection (such as codes or algorithms used to identify subjects) should be listed in detail.”, some statements are not applicable to all settings, like “Explain how the study size was arrived at”, as we have collected all the available data of the considered period of time, not a sample.

Please, if any other important RECORD statement was not met, we will be glad to revise.

1.1 In particular, the covariates used in the primary analysis are not well described and are very hard to understand. ANSWER: We have added a better explanation to the new version of the manuscript providing details on the covariates included.

Please, if any details about the covariates is still not clear, we will be glad to revise.

1.2 How is ethnicity defined in hospital collections in Brazil? ANSWER: We have revised the manuscript methods, including a better explanation of how ethnicity was defined in our Study:

Ethnicity was based on patient self-declaration of race/color at the time of admission, which could be classified as: white, black, brown, yellow, native Brazilian or not informed

1.3 How is this data accessed? ANSWER: As stated in methods section, the data is available at the SIHSUS repository (ftp://ftp.datasus.gov.br/dissemin/publicos/SIHSUS/200801_/Dados). This repository was accessed with R microdatasus package. Thus, we have revised the text in order to make clear how the data was accessed:

All analyses were based on hospitalization authorization (AIH), which is public data available at the SIHSUS repository (ftp://ftp.datasus.gov.br/dissemin/publicos/SIHSUS/200801_/Dados) until the end of January 2021. In the AIH database, each hospitalization receives a unique key called the AIH number. If necessary, duplicated AIH numbers were filtered, considering only the main hospitalization record.

In addition to cleaning and manipulation process in R language, the data was accessed with microdatasus package13. The programming code, dictionary and deidentified admissions data used in this study can be found at a public repository (https://github.com).

1.4 What comorbidities were included, how were they identified (ICD codes?), does this leave room for error and which errors if so? ANSWER: Comorbidities were included based on associated ICD-10 codes, which were fulfilled by hospitals in order to require reimbursement from the Ministry of Health. As comorbidities refer to secondary-diagnosis, this information is not mandatory (just primary-diagnosis is mandatory for means of reimbursement in the public health system in Brazil). On this basis, comorbidities tend to be under reported. All ICD-10 codes are available in Table S1 in the supplementary material. 

1.5 I would suggest the authors carefully go through not just the STROBE checklist, but the accompanying papers to ensure that they are indeed meeting the guidelines for reporting. ANSWER: We have revised our manuscript in order to be consistent with RECORD statements checklist (http://record-statement.org). Please, if any other important RECORD statement was not met, we will be glad to review.

2 There is no information on how deaths were garnered from the records, which is a major weakness of the research. Were these in-hospital deaths? Does Brazil have a linked hospital/death reporting system? If these are in-hospital deaths, reporting lags may be a large issue - if not, death reporting across regions should be considered as a potential weakness. ANSWER: There is no public linked data with National Death Information System (SIM), but there is a particular field in hospitalization records stating if it was a discharge due to death or not. Thus, the deaths here reported regard only to in-hospital deaths. As we are only dealing with in-hospital deaths based on discharge data from the year 2020, we believe that reporting lag is not a major issue. Nevertheless, we agree with reviewer’s points and the revised manuscript states clear how deaths were garnered from the records and also emphasize the limitation of only dealing with in-hospital deaths in discussion section.

The death outcome was based on the discharge information field (discharge due to death) available in the hospitalization records, thus covering only the in-hospital deaths. 

[…]

Finally, we highlight that our data only covers in-hospital deaths reimbursed by SUS and should not be generalized to hospitalizations or deaths in other settings as home deaths.

3 One hypothesis for the changing hospital CFR in many places is the nature of the pandemic itself. In the UK, hospital CFRs fell from the peak until the second wave, and then increased again. This is potentially due to an overwhelming effect, whereby a large number of COVID-19 cases changes the type of patient who is admitted to hospital, and thus changes the denominator for the in-hospital CFR. While it may be impossible to fully examine the impact of such changes, it is clear from the trends in CFR when stratified by age and ethnicity that there is some impact. It might be useful for this study to look at the hospital CFR for each region over time against the current caseload of that area, although this is an addition that would take some extra work. ANSWER: We agree with the comment. All our trend estimates are presented now in an age-standardized fashion, which deals with potential changes in the age structure of the population affected during the wave. A brief explanation was included in methods section:

In order to make group comparisons, the weekly hospital CRF estimates were standardized through direct method assuming the age pattern of the total number of hospital admissions in the entire period as the reference population.

In addition, we have added a correlation analysis of the age-standardized CRF against the week caseload of each region. These results were included in the manuscript results in the Figure S3. A brief explanation was also included in methods section:

Finally, we also planned to run additional Spearman correlation analysis to evaluate the effect of local caseload against the hospital CFR.

Thanks to the comment, we think we have more robust estimates now.

4 There is insufficient information in the text about missing data. While routine hospital data is incredibly useful, it is also usually filled with missing fields. While the dataset appears to be impressive, there should be a detailed discussion in the manuscript of how missing data was managed. ANSWER: All the major fields included in the analyses are obligatory fields. Nevertheless, we identified potential missing information in ethnicity (which was describe in results section, where about 20% of the data was stated as “not informed”) and comorbidities field, which could suffer from underreporting. 

Because of the limited amplitude of the number of variables impacted and their profile (ethnicity and comorbidities), we considered that imputation or other missing values procedures were not applicable.

Still, we have revised our discussion section and highlighted the statement about underreporting:

First, in addition to a high level of missing information on ethnicity, epidemiological information on pre-existing comorbidities, have low quality and completeness in administrative datasets since this information is not mandatory for reimbursement. Based on this, our estimates from these variables must be interpreted with caution due to potential underreporting bias. 

5 The statistical analysis is currently not fully described. The method of obtaining smoothed curves should be elucidated. ANSWER: We have revised the methods section and a better description of the method of obtaining smoothed curves is now available:

Then, smoothing time series plots were created to analyse the trends of the hospital case-fatality rate and its variation according to sex, age, ethnicity, length of hospital stay and ICU use. Smoothing was based on local polynomial regression fitting (loess)14 method available in geom_smooth() R function, which fits a polynomial surface determined by one or more numerical predictors using local fitting (fit is made using points in each point neighbourhood weighted by their distance). Finally, we also planned to run additional Spearman correlation analysis to evaluate the effect of local caseload against the hospital CFR.

6 A somewhat minor point, but to me the tables are extremely hard to read. ANSWER: Thanks. We have revised all the tables to make them clearer.

6.1 I would suggest having more columns and fewer rows, perhaps breaking each table down by age group. ANSWER: Thanks. Although not adding more columns and fewer rows, we have revised all the tables to make them clearer.

6.2 Similarly, the regression outputs are hard to read, especially given that the reference categories have been excluded. ANSWER: Thanks. We have revised all the tables to make them clearer, including the reference categories.

7 While the introduction and discussion are good, I would ask for more information particularly in the introduction about the Brazilian hospital system. For international readers, there is scant detail on how it works - an additional paragraph would be very helpful to understand the context. ANSWER: We have expanded the description of the SIHSUS in the "Study Setting" section:

"SUS provides health care, free of charge at the point of service, to the entire Brazilian population, covering both ambulatory and hospital care. The reimbursement of hospitalizations by SUS budget is done through the hospital admission authorizations (Autorização de Internação Hospitalar, AIH), document that identify the patient and the services performed during the hospital stay. The AIH is generated at the time o of admission to public or private hospitals that provide services for SUS, and are sent monthly to the Ministry of Health to enable billing of services delivered. The AIH are grouped and managed through the Hospital Information System (Sistema de Informações Hospitalares do SUS, SIHSUS), an administrative system that supports planning, regulation and control. Besides, SIHSUS allows the assessment of hospital morbidity and mortality profile and the quality of health care offered to the population, providing elements to improve health policies."

8 Currently, the link the authors have provided to their data/code goes to github's main webpage. Probably a typo. ANSWER: We have not stated the specific github repository in the text for anonymity purpose during the peer review process, but we will be glad to share the complete code if wanted or we can already describe the full link if allowed by editors.

 

No. Reviewer’s #2 Comment Authors’ Answer

-- This article describes the evolution of hospital lethality due to COVID-19 in the public hospital care system in Brazil. The article uses patient health data collected by the Brazilian public social insurance system, which covers more than 300,000 patients, with the indication of the duration of hospitalization and the outcome of the hospitalization. This large dataset allows a precise analysis of the evolution of hospital lethality and the article is of undeniable interest ANSWER: Thank you very much. We carefully considered each of the suggestions in order to enhance the clarity and the quality of our paper.

1 Brazil is a very big country and have a very heterogeneous geography. The authors could present in more detail the differences between the different States of Brazil (instead regions) ANSWER: We agree with this comment. Although not being able in the scope of this paper to run all the analyses separately for each Brazilian state, we have now included a supplementary analysis of the age-standardized hospital CFR timeline on each of the 27 Brazilian states. This can be found in Figure S3.

2 It would be interesting to have indication on the duration of the hospitalization (statistical distribution, relationship with the outcome of the disease, etc.). ANSWER: Thanks for the comment. We have revised the entire structure of Table 1, including this information.

In addition, we believe that important relationships have been addressed in the CFR according to the length of stay and, icu need in the survival analysis model.

3 To study the dynamics and evolution of CFR and to compare rates over time, if we want to exclude known factors related to death (mainly age), we have to standardize on age between weeks: therefore, we have to take into account the evolution of the age structure of patients over time. ANSWER: Thanks for the comment. All our trend estimates are presented now in an age-standardized fashion, which deals with potential changes in the age structure of the population affected during the wave. A brief explanation was included in methods section:

In order to make group comparisons, the weekly hospital CRF estimates were standardized through direct method assuming the age pattern of the total number of hospital admissions in the entire period as the reference population.

We think we have more robust estimates now.

4 In several countries a strong correlation between morbidity and case fatality rates has been observed. It would be interesting to have this analysis also for Brazil. And in particular to do it by state, because the differences between regions in the evolution of the disease are sensitive (but as mentioned before, when comparing geographical units, it is necessary to standardize on age). ANSWER: Thanks for the comment. Following the suggestion, we sought to analyze the direct impact of morbidity on CFR through the correlation between the number of admissions per week and the age-standardized CFR for each region. So, we have added a correlation analysis of the age-standardized CRF against the week caseload of each region. A brief explanation was included in methods section:

Finally, we also planned to run additional Spearman correlation analysis to evaluate the effect of local caseload against the hospital CFR.

We identified a strong correlation in two of the 5 Brazilian regions. We have included this data in the revised manuscript and in the supplementary material Figure S3. Although not being able in the scope of this paper to run all the analyses separately for each of the 27 Brazilian states, we have also included a supplementary analysis of the age-standardized hospital CFR timeline on each state. This can be found in Figure S2

Thanks to the comment, we think we have more robust estimates now.

5 Finally, it should be noted that it is always difficult to compare hospital case fatality rates between countries, because even within areas with comparable health systems (e.g., the EU), these hospital case fatality rates show differences that cannot be explained solely by differences in patient management, but first by the difference between countries in definition, declaration and reporting systems for morbidity and mortality ANSWER: We totally agree with the reviewer’s point of view. When we brought evidence from other countries, our idea was to highlight the “trend” and not the “value of case-fatalily rates" per se. To make it clear, we insert new sentences in the discussion.

It is worth noting that case fatality rates are not directly comparable among countries due to high heterogeneity in terms of countries’ health information systems (in-hospital versus 30-day after discharge), quality of registry (mandatory versus voluntary report), disease classification (ICD-10 versus ICD-9 or other earlier versions), and completeness of the information.

We totally agree with the reviewer. 

6 The quality of illustration can be improved. Maps are welcome. ANSWER: Thanks. We have revised our Figures and included more illustrations in the supplementary material

We have all the Figures in high quality vectorial files (SVG), but the PLOS One system doesn’t recognize it as a “Figure”. Thus, in addition to the .tif files, we have uploaded a compressed file with all figures in high quality vectorial format (SVG).

MR1 Why calculate the lethality rate per week, when the calculation can be done per day and then smoothed per week? ANSWER: When dealing with daily hospitalizations instead of weekly there is a high potential for outliers (days with high or low rates), mainly in the first epidemic weeks. The epidemic week is standardized method that performs better and allows the comparison of data year after year. Similar to previous studies and consistent with WHO recommendations on dissemination on epidemiological information on cases and outbreaks of diseases under the International Health Regulations, we think that the epidemic week fashion is more stable.

MR2 Line 58, 59. Hospital (or inpatient, or intensive care unit inpatient) fatality rate (not mortality rate) ANSWER: Ok, revised

MR3 Line 63-71: need to adjust the terminology (death rates, hospital fatality rate, adjusted mortality…). I think that « inpatient case-fatality rate » or « hospital case-fatality rate » is appropriate. ANSWER: Ok, the term hospital case-fatality rate was adopted

MR4 Line 142: better “proportion between the number of COVID-19 related admission that evolved to death and the total number of COVID-19 admission in that week” ANSWER: Ok, revised

MR5 Line 170-179: all these results by category (age, ethnicity, comorb.) in the hospitalized COVID-19 population must be compared with the proportion of the same category in the global population, and the authors must indicate if the differences are statistically significant. Table 1 must present these results. ANSWER: Thanks for the comment. Nevertheless, we would like to emphasize that the entire sample was hospitalized, and it is not possible to calculate the mentioned proportion. Though, we worked on a clearer version of the data description, including the proportion of hospitalizations according to each category and the occurrence or not of death in Table 1.

MR6 Line 182-184: Table 1 is mixing different information. I think it must be splitled in various tables. For example, Comorbidity analysis with CFR differences will be very interesting. ANSWER: We have worked on a clearer version of the data description in Table 1 including the proportion of hospitalizations according to each category and the occurrence or not of death. 

Nevertheless, a better analysis of the CRF according to the presence of comorbidities and other factors is also presented in Table 2 with Cox regression results.

MR7 Line 195-201: Must present morbidity rates and not only morbidity. ANSWER: We agree and now we have presented morbidity rates as well in the text.

MR8 Line 208-212: comparison of CFR between ethnic group is interesting only if major cause of death (age) in excluded: data must be standardized on age before comparison. Also, size of groups is different: it is needed to present confidence intervals. ANSWER: Thanks for the comment. We have revised our trend estimates in an age-standardized fashion, which deals with potential changes in the age structure of the population affected during the wave.

We think we have more robust estimates now. In addition, confidence intervals for the cited estimates were also included in the text.

MR9 Line 223-226: Is there a relationship between morbidity and CFR, as observed in other countries? Globally? By region? It is possible that changes in CFR are directly related to morbidity, so the study should be refined by analyzing data on regions (or better on states). ANSWER: Thanks for the comment. Following the suggestion, we sought to analyze the direct impact of morbidity on CFR through the correlation between the number of admissions per week and the age-standardized CFR for each region. We identified a strong correlation in two of the 5 Brazilian regions. We have included this data in the revised manuscript and a figure about it in the supplementary material Figure S3. 

Although not being able in the scope of this paper to run all the analyses separately for each Brazilian state, we have also included a supplementary analysis of the age-standardized hospital CFR timeline on each of the 27 Brazilian states. This can be found in Figure S3

MR10 Line 235-239: this is very strange. Perhaps the explanations given in the discussion (data quality) could be the subject of an earlier paragraph in the data and methods section. ANSWER: Exactly. As mentioned in the discussion section, we believe that this is a potential limitation. Our database is very valid in some points such as the presence of the discharge outcome, however, it is still deficient in other aspects such as the description of comorbidities.

MR11 Line 246: there is no indication of the source of the data on the health care system or the criteria used to characterize it (“well-equipped and staffed hospital” is not enough). ANSWER: Thanks. We have revised this definition. This field is classified by the Ministry of Health according to the set of procedures reimbursed. The high complexity is the set of procedures that, in the context of SUS, involve high technology and high cost.

We have added this definition in the TableS1 in Supplementary information.

MR12 Line 266: reference needed. ANSWER: Thanks. I’m afraid that the all the references were included at the end of the whole sentence. Please, if there is a better way of citing this statement, we will be glad to revise.

MR13 Line 336: remark: the same level of CFR was observed in France in second and third wave. ANSWER: Thanks for the comment

Notes: MR: Minor Remark

 

No. Reviewer’s Comments Authors’ Answers

 Reviewer #3 

1 This is a straightforward epidemiological study of a cohort of nationwide COVID-admissions in Brazil from March to October 2020 analyzing in-hospital mortality and its associated factors. The study has been well planned and executed. The sheer size of the cohort gives the opportunity to examine factors influencing the CFR with a high power. ANSWER: Thank you very much. We considered carefully each of the suggestions in order to enhance the clarity and the quality of our paper.

2 The decrease in CFR is highest and most pronounced in the short term hospitalized group. This raises the question, whether the populations hospitalized in different categories (short term, non-ICU vs. ICU) changed over time. Examining these questions could be important for the interpretation of the time trends shown. ANSWER: We would like to thank the reviewer for the comment. Indeed, the population structure could have changed overtime and this could be observed in crude estimates.

Thus, we have revised our trend estimates and now they are presented in an age-standardized fashion, which deals with potential changes in the age structure of the population affected during the wave. A brief explanation was included in methods section:

In order to make group comparisons, the weekly hospital CRF estimates were standardized through direct method assuming the age pattern of the total number of hospital admissions in the entire period as the reference population.

We think we have more robust estimates now.

3 In addition, if there are changes in the populations an additional analysis stratified for variables with clear changes over time should be done (or reported if already done). ANSWER: Thanks for the comment. We have revised our trend estimates in an age-standardized fashion, which deals with potential changes in the age structure of the population affected during the wave.

We think we have more robust estimates now.

---

## [Editor Report · Decision Letter 1]

1 Jul 2021

Trends in COVID-19 case-fatality rates in Brazilian public hospitals: a longitudinal cohort of 398,063 hospital admissions from 1st March to 3rd October 2020

PONE-D-21-11421R1

Dear Dr. Zimmermann,

We’re pleased to inform you that your manuscript has been judged scientifically suitable for publication and will be formally accepted for publication once it meets all outstanding technical requirements.

Kind regards,

Aleksandar R. Zivkovic

Academic Editor

PLOS ONE

---

## [Editor Report · Acceptance letter]

6 Jul 2021

PONE-D-21-11421R1 

Trends in COVID-19 case-fatality rates in Brazilian public hospitals: a longitudinal cohort of 398,063 hospital admissions from 1st March to 3rd October 2020 

Dear Dr. Zimmermann:

I'm pleased to inform you that your manuscript has been deemed suitable for publication in PLOS ONE. Congratulations! Your manuscript is now with our production department. 

Kind regards, 

on behalf of

Dr. Aleksandar R. Zivkovic 

Academic Editor

PLOS ONE